# Preferential Elimination of Ba²⁺ through Irreversible Biogenic Manganese Oxide Sequestration

**Yukinori Tani** [1,2,*], **Satomi Kakinuma** [1], **Jianing Chang** [2], **Kazuya Tanaka** [3] **and Naoyuki Miyata** [4]

[1] Department of Environmental and Life Sciences, School of Food and Nutritional Sciences, University of Shizuoka, 52-1 Yada, Shizuoka 422-8526, Japan; st.kakinuma@gmail.com

[2] Department of Environmental Health Sciences, Graduate School of Nutritional and Environmental Sciences, University of Shizuoka, 52-1 Yada, Shizuoka 422-8526, Japan; jianing_ch0802@sina.com

[3] Advanced Science Research Center, Japan Atomic Energy Agency, Tokai, Ibaraki 319-1195, Japan; tanaka.kazuya@jaea.go.jp

[4] Department of Biological Environment, Akita Prefectural University, Shimoshinjo-Nakano, Akita 010-0195, Japan; nmiyata@akita-pu.ac.jp

* Correspondence: taniy@u-shizuoka-ken.ac.jp; Tel.: +81-54-264-5797

**Abstract:** Biogenic manganese oxides (BMOs) formed in a culture of the Mn(II)-oxidizing fungus *Acremonium strictum* strain KR21-2 are known to retain enzymatic Mn(II) oxidation activity. Consequently, these are increasingly attracting attention as a substrate for eliminating toxic elements from contaminated wastewaters. In this study, we examined the Ba²⁺ sequestration potential of enzymatically active BMOs with and without exogenous Mn²⁺. The BMOs readily oxidized exogenous Mn²⁺ to produce another BMO phase, and subsequently sequestered Ba²⁺ at a pH of 7.0, with irreversible Ba²⁺ sequestration as the dominant pathway. Extended X-ray absorption fine structure spectroscopy and X-ray diffraction analyses demonstrated alteration from turbostratic to tightly stacked birnessite through possible Ba²⁺ incorporation into the interlayer. The irreversible sequestration of Sr²⁺, Ca²⁺, and Mg²⁺ was insignificant, and the turbostratic birnessite structure was preserved. Results from competitive sequestration experiments revealed that the BMOs favored Ba²⁺ over Sr²⁺, Ca²⁺, and Mg²⁺. These results explain the preferential accumulation of Ba²⁺ in natural Mn oxide phases produced by microbes under circumneutral environmental conditions. These findings highlight the potential for applying enzymatically active BMOs for eliminating Ba²⁺ from contaminated wastewaters.

**Keywords:** biogenic manganese oxide; Mn(II) oxidizing fungi; sequestration of barium(II) ion; *Acrenonium strictum*; birnessite

## 1. Introduction

Barium is a toxic alkaline earth metal [1,2], and Ba²⁺ levels have been elevated in many aquatic environments by anthropogenic activities such as mining [3], coal seam gas utilization [4], and shale gas extraction [5,6]. These enhanced Ba levels threaten humans and ecosystems worldwide. Therefore, developing a cost-effective Ba²⁺ remediation systems requires urgent attention.

In aquatic and terrestrial environments, manganese (Mn) oxide phases readily accumulate Ba²⁺ and other heavy metal ions such as Ni²⁺, Co²⁺, and Zn²⁺. The preferential accommodation of Ba²⁺ in the tunnels of tectomanganates such as hollandite (2 × 2) and romanechite (2 × 3) partly explains Ba²⁺ accumulation in natural Mn oxide phases [7–9]. Phyllomanganates such as birnessite and buserite (vernadite) also accumulate Ba²⁺ under certain environmental conditions [10–15], although the underlying mechanisms for Ba²⁺ accumulation in manganese oxide phases remain uncertain.

Under circumneutral pH conditions, bacterial and fungal Mn oxide formation enzymatically proceeds faster than heterogeneous Mn(II) oxidation catalyzed by mineral

surfaces [16–18]. The formation of biogenic Mn oxides (BMOs) serves in scavenging heavy metal cations including $Zn^{2+}$, $Ni^{2+}$, $Co^{2+}$, and $Pb^{2+}$ from aquatic environments because of the high sequestration affinity and capacity of BMOs [19–28].

According to previous studies [29–31], fungal BMOs produced by *Acremonium strictum* KR21-2 maintain the activity of an Mn(II)-oxidizing enzyme in the oxide phase, and effectively oxidize exogenous $Mn^{2+}$ to form another BMO phase. This process significantly enhances the efficiency of heavy metal [32–35] and rare-earth metal [36,37] sequestration by providing new sorption sites and minimizing competition for exogenous $Mn^{2+}$ sorption. In addition, enzymatically active BMOs improve the indirect oxidation efficiencies of As(III) to As(V) [30], Co(II) to Co(III) [33], and Cr(III) to Cr(VI) [38] through continuous reoxidation of reduced $Mn^{2+}$, which is one of the causes of surface passivation. Consequently, in addition to the high sequestration capacity and oxidizing ability, enzymatically active BMOs exhibit a potential for a continuous remediation of contaminated wastewaters as well as metal recovery.

The aims of this study were to examine the $Ba^{2+}$ sequestration process associated with enzymatic BMO formation and to elucidate factors for the preferential accumulation of this ion in Mn oxide phases in the environment. The BMO alteration linked to the $Ba^{2+}$ sequestration is also discussed relative to the sequestration reversibility and selectivity for alkali earth metal ions such as $Sr^{2+}$, $Ca^{2+}$, and $Mg^{2+}$. The results of this study demonstrate the potential application of enzymatically active BMOs for $Ba^{2+}$ removal from contaminated wastewaters.

## 2. Materials and Methods

*A. strictum* KR21-2, which enzymatically oxidizes Mn(II) to BMOs [39–41], was incubated at 25 °C in a HAY liquid medium (pH of 7.0) supplemented with 1 mM $Mn^{2+}$, as described previously [29,32–34], slightly modified by using $Mn(NO_3)_2$ instead of $MnSO_4$. After 72 h of incubation, BMOs with fungal mycelia were harvested and washed thrice with 20 mM 4-(2-hydroxyethyl)–1–piperazineethanesulphonic acid (HEPES) buffer (pH of 7.0 adjusted using NaOH). These served as the "newly formed BMOs" for $Ba^{2+}$ sequestration experiments within 1 h of washing (denoted as "newly formed BMO").

In the sequestration experiments, all metal ions involved ($Mn^{2+}$, $Ba^{2+}$, $Sr^{2+}$, $Ca^{2+}$, and $Mg^{2+}$) were as nitrate salts because $Ba^{2+}$ readily precipitates with $SO_4^{2-}$ (the solubility product, $K_{sp}$, of $BaSO_4$ is $10^{-9.97}$ [42]). The enzymatically active newly formed BMOs (1 mM as Mn) were mixed with 0–10 mM $Ba(NO_3)_2$ with or without 1 mM $Mn(NO_3)_2$ in 20 mM HEPES buffer (50 mL) at a pH of 7.0 (adjusted using NaOH) under air-equilibrated (aerobic) conditions at 25 °C on a reciprocal shaker at 105 strokes·$min^{-1}$ (NR–10, Taitec, Nagoya, Aichi, Japan). To maintain aerobic conditions, we used 100 mL Erlenmeyer flasks with cotton stoppers. This procedure was performed thrice, with the bathing solution renewed every 24 h. To elucidate the effects of the Mn(II) oxidase activity in the BMOs, we inactivated the associated Mn(II) oxidase by heating the newly formed BMOs for 2 h in a water bath (Thermo Minder Mini-80, Taitec, Nagoya, Aichi, Japan) at 85 °C [29], followed by cooling of the samples to room temperature at around 20 °C (denoted as "heated BMO" hereafter). These cooled samples were collected, washed thrice with a 20 mM HEPES buffer at pH of 7.0, and used in the $Ba^{2+}$ sequestration experiments under aerobic conditions. To compare the sequestration properties of the alkaline earth metal ions, we also conducted experiments with solutions of $Sr^{2+}$, $Ca^{2+}$, and $Mg^{2+}$ in the 20 mM HEPES buffer. For competitive sequestration experiments, newly formed BMOs were treated thrice in mixed solutions of $Ba^{2+}$ with $Sr^{2+}$, $Ca^{2+}$, or $Mg^{2+}$, and with or without exogenous $Mn^{2+}$. In all sequestration experiments, supernatants were sampled at 0, 2, 8, 16, and 24 h for each treatment and separated using a centrifugal filter unit (Durapore PVDF 0.1 μm, Merk Millipore, Burlington, MA, USA; 12,000× *g* for 2 min). The dissolved metal concentrations of the supernatants were measured using a 730-ES inductivity coupled plasma atomic emission spectrometer (ICP-AES, Agilent Technology, Santa Clara, CA, USA).

The two-step extraction protocol in this study involved using aqueous 10 mM $Cu(NO_3)_2$ (pH of 4.8) and 50 mM hydroxylamine hydrochloride for speciation of the $Ba^{2+}$ and $Mn^{2+}$ sequestered by the BMOs, as described previously [29,32–34]. This extraction sequence commonly serves for fractionating adsorbed Mn(II) and oxidized Mn from BMOs [43–45]. The $Ba^{2+}$ and other alkaline earth metal ion fractions dissolved in aqueous $Cu(NO_3)_2$, and the subsequent hydroxylamine hydrochloride extracts were termed as exchangeable (reversible) and reducible (irreversible) fractions, respectively. The metal concentrations in the extracts were also determined using ICP-AES after dilution with 1.0 M $HNO_3$. The total metal ions extracted via this two-step extraction sequence is hereafter referred to as "solid". All the sequestration and extraction experiments were conducted in triplicate (*n* = 3), and data in the figures and tables are shown as mean ± standard deviation.

X-ray diffraction (XRD) measurements were performed for the BMOs using a Rigaku Rint2500 diffractometer (Akishima, Tokyo, Japan) involving CuK$\alpha$ radiation at 26 mA and 40 kV. Lyophilized BMO samples were placed on a glass holder and scanned over a 2$\theta$ range of 5–70° at 1.0° min$^{-1}$ using a 0.02° step interval. The diffractograms were smoothed using a 10-point moving average to enhance the display of broad peaks.

Manganese K-edge extended X-ray absorption fine structure (EXAFS) data for the BMO samples were obtained at the BL12C in the Photon Factory, KEK (Tsukuba, Japan). Lyophilized BMO samples were diluted and adequately mixed with boron nitride (BN). After homogenization, the mixed BMO–BN powders were pressed into discs of appropriate thicknesses for EXAFS measurements in the transmission mode. The intensities of the incident and transmitted X-rays were monitored at room temperature using ionization chambers. Conversely, Barium K-edge EXAFS data for the BMOs treated with $Ba^{2+}$ were measured at the BL01B1 in the SPring-8 facility (Hyogo, Japan). The lyophilized samples for the Ba-edge EXAFS were also pressed to form discs, without BN dilution. The EXAFS data were also generated in the transmission mode using ionization chambers and analyzed using ver. 2.5.9 of REX2000 (Rigaku Co. Ltd., Akishima, Tokyo, Japan).

## 3. Results and Discussion

### 3.1. Exogenous $Mn^{2+}$ Oxidation by Newly Formed BMOs

Under aerated (air-equilibrated) conditions, the newly formed BMOs (1 mM Mn) readily converted 1 mM exogenous $Mn^{2+}$ to solid phase Mn in 20 mM HEPES at a pH of 7.0 (cumulative sequestration efficiency > 98.7 ± 0.1%) and subsequently produced another solid phase (Figure S1A and Table S1). Two-step extraction experiments confirmed that these solid phases mainly comprised reducible (oxidized) Mn (>84.2 ± 0.1%) with minor (<15.8 ± 0.1%) exchangeable $Mn^{2+}$ after every 24 h in the repeated treatment (Figure S1B). The XRD patterns of the newly formed BMOs are characterized by broad peaks for the (001) and (002) basal reflections at ~7.4 and ~3.6 Å, respectively (Figure S2a), indicating a turbostratic birnessite structure [45]. These patterns were maintained even after repeated exogenous $Mn^{2+}$ oxidation (Figure S2b–d).

### 3.2. $Ba^{2+}$ Sequestration by Newly Formed or Heated BMOs with Exogenous $Mn^{2+}$

After adding newly formed BMOs (1 mM as Mn) to a mixture of 1 mM $Mn(NO_3)_2$ and 1 mM $Ba(NO_3)_2$ (20 mM HEPES at a pH of 7.0), the exogenous $Mn^{2+}$ concentrations decreased over time, with >99% subsequently converted to solid Mn upon termination of each treatment (Figure 1A and Table S1). Two-step extraction data also revealed that oxidized (reducible) Mn dominated the solid Mn phase throughout the repeated treatment (93.4 ± 0.1% to 94.7 ± 0.5%) (Figure 1D), indicating active Mn oxide formation by the newly formed BMOs. In fact, dissolved $Ba^{2+}$ was efficiently sequestered, with the content reducing from 27.0 ± 0.5% upon the initial treatment to 10.0 ± 0.3% after the third treatment (Figure 1A). The cumulative $Ba^{2+}$ concentration increased by up to 0.45 ± 0.00 mM (Figure 1C) (the cumulative efficiency was 16.1 ± 0.0%, Table S1). The molar ratio of the sequestered $Ba^{2+}$ relative to the oxidized Mn ($Ba^{2+}_{seq}/Mn_{oxide}$) was 11.9 ± 0.1 mol % at the end of the repeated treatment.

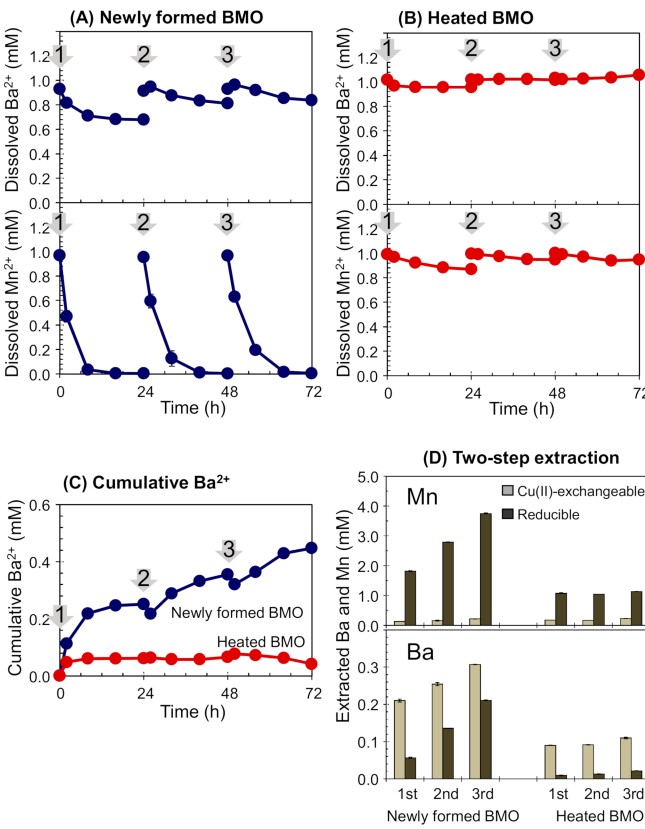

**Figure 1.** Illustration of the repeated treatment of the (**A**) newly formed and (**B**) heated biogenic manganese oxides (1 mM Mn) with mixtures of 1 mM $Ba(NO_3)_2$ and 1 mM $Mn(NO_3)_2$ in 20 mM 4-(2-hydroxyethyl)–1–piperazineethanesulphonic acid (HEPES) (pH of 7.0). (**C**) Cumulative concentration of the sequestered $Ba^{2+}$ and (**D**) exchangeable and reducible Ba and Mn in the solid phases, assessed via the two-step extraction. Bathing solutions were renewed every 24 h (indicated by arrows).

In contrast, upon treatment with heated BMOs, the exogenous $Mn^{2+}$ slightly reduced (Figure 1B), thereby minimally increasing the solid Mn phase (Figure 1D). This behavior is attributed to the lack of enzymatic Mn(II) oxidation ability of the heated BMOs [29]. The dissolved $Ba^{2+}$ concentration also slightly decreased, producing a minor cumulative $Ba^{2+}$ sequestration of 0.04 ± 0.01 mM (efficiency ≈ 1.4%; Figure 1C). Here, a significantly lower $Ba^{2+}_{seq}/Mn_{oxide}$ ratio (3.7 ± 0.5 mol %) was obtained, indicating competitive sorption of unreacted exogenous $Mn^{2+}$ (Figure 1B) and $Ba^{2+}$ on the BMO surface, as previously demonstrated for heavy metal ion sequestration [32,34]. Consequently, the enzymatic $Mn^{2+}$ oxidation ability enhanced the $Ba^{2+}$ sequestration efficiency not only by preparing new accommodation sites, for example, a new BMO phase, but also by minimizing the impact of exogenous $Mn^{2+}$ as a sorption competitor. In fact, newly formed BMOs without exogenous $Mn^{2+}$ produced the highest $Ba^{2+}_{seq}/Mn_{oxide}$ ratio of 19.5 ± 0.9 mol %, with the cumulative sequestered $Ba^{2+}$ concentration limited to 0.19 ± 0.01 mM (Figure S3 and Table S1), which is significantly lower than that with 1 mM exogenous $Mn^{2+}$ (0.45 ± 0.00 mM; Figure 1C).

Interestingly, the two-step extraction data revealed $Ba^{2+}$ sequestration reversibility differences between the newly formed and heated BMOs. In the newly formed BMOs, the total sequestered $Ba^{2+}$ contained up to 40.7 ± 0.1% irreversible $Ba^{2+}$ (extracted as the reducible phase) (Figure 1D and Table S1), whereas for the heated BMOs, the sequestered $Ba^{2+}$ was mainly extracted as exchangeable $Ba^{2+}$ (90.3 ± 0.9 to 83.6 ± 0.0%) during the repeated treatment (Figure 1D). Similar trends were observed for initial $Ba^{2+}$ concentrations ranging from 0.15 to 10 mM and 1 mM exogenous $Mn^{2+}$ (Figure 2). In fact, at initial $Ba^{2+}$ concentrations of 0.15, 3, and 10 mM, irreversible $Ba^{2+}$ represents 58.3 ± 1.2%, 44.1 ± 0.6%, and 44.4 ± 0.6% sequestration on the newly formed BMOs, respectively (Figure 2 and

Table S1). However, for the $Ba^{2+}$ sequestered by the heated BMOs, exchangeable $Ba^{2+}$ makes up $76.3 \pm 0.5\%$, $82.1 \pm 0.5\%$, and $82.5 \pm 0.4\%$ for corresponding initial $Ba^{2+}$ concentrations (Figure 2 and Table S1). In addition, even for the newly formed BMOs, irreversible $Ba^{2+}$ incorporation is scarce without exogenous $Mn^{2+}$ addition, with >93% of the sequestered $Ba^{2+}$ as exchangeable $Ba^{2+}$ (Figure S3D).

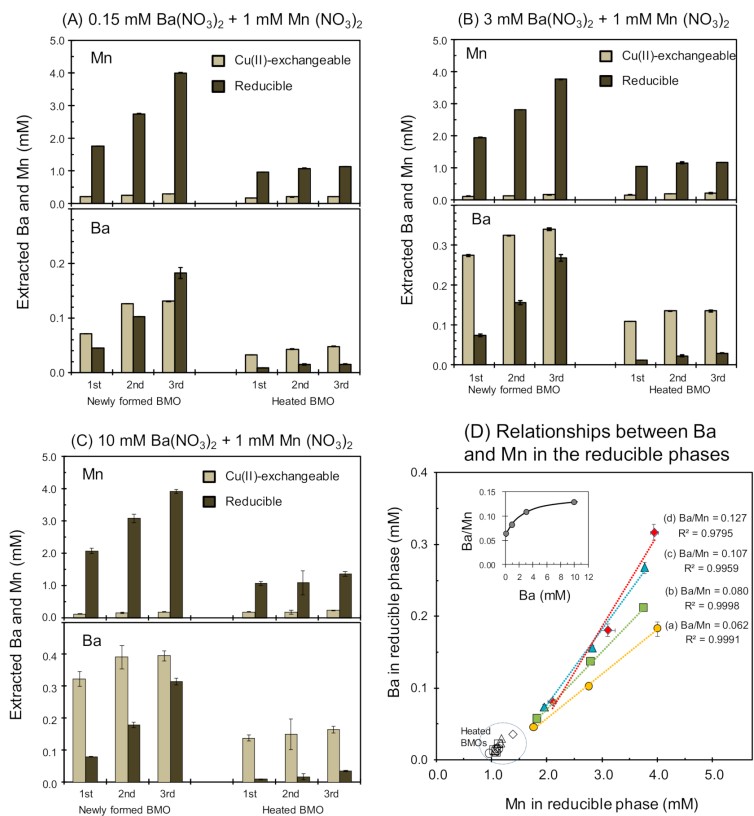

**Figure 2.** Diagram showing the two-step extraction of $Ba^{2+}$ and Mn from the newly formed and heated biogenic manganese oxides through repeated treatment with mixtures of (**A**) 0.15 mM, (**B**) 3 mM, and (**C**) 10 mM $Ba(NO_3)_2$ with 1 mM $Mn(NO_3)_2$ in 20 mM HEPES (pH of 7.0). (**D**) Plot displaying linear relationships between the extracted Ba and Mn in reducible phases, with the inset showing the Ba/Mn molar ratios as a function of the initial $Ba^{2+}$ concentration.

Linear correlations between the amounts of irreversible $Ba^{2+}$ and reducible (oxidized) Mn in the solid phase ($R^2 > 0.97$) were observed throughout the repeated treatments (Figure 2D). From the slopes of the linear relationship curves, molar ratios of the irreversible $Ba^{2+}$ to the additional oxidized Mn phase from the exogenous $Mn^{2+}$ increased from 0.062, 0.080, and 0.107 to 0.127 as the initial $Ba^{2+}$ concentrations increased from 0.15, 1, and 3 to 10 mM, respectively (Figure 2D inset). Considering the incorporation of irreversible $Ba^{2+}$ into the additional Mn oxide phase, 16.2, 12.5, 9.4, and 7.9 moles of oxidized Mn accommodated 1 mole of irreversible $Ba^{2+}$, on average, as the initial $Ba^{2+}$ concentrations changed from 0.15, 1, and 3 to 10 mM, respectively. Although isomorphic substitution with structural $Mn^{4+}$ is reported to cause irreversible sequestration of $Ni^{2+}$ [34], this is impossible for $Ba^{2+}$ because of its significantly higher ionic radius (1.49–1.75 Å [46]) compared to that of $Mn^{4+}$ (0.53–0.67 Å [46]).

### 3.3. BMO Alteration from Turbostratic to Tightly Stacked Birnessite

The Mn K-edge EXAFS data for the newly formed BMOs (untreated) were similar to those of chemically synthesized $\delta$-$MnO_2$ (Figure 3). This is consistent with the fact that the original BMOs were turbostratic analogues of birnessite [47]. Even after adding the exogenous $Mn^{2+}$, the newly formed BMOs maintained the EXAFS oscillations throughout

the repeated treatment in 10 mM $Ba^{2+}$ (Figure 3). This indicated no remarkable alteration in the structural alignment of Mn in the resultant BMOs, although the $Ba^{2+}$ sequestration reversibility largely became irreversible. This behavior is inconsistent with the formation of tectomanganates such as hollandite (2 × 2) and romanechite (2 × 3). These naturally occurring Mn oxides are considered the most suitable for accommodating $Ba^{2+}$ because of their tunnel structures [7,9,11,28]. Therefore, we inferred that under the experimental conditions in this study, the coexisting $Ba^{2+}$ failed to directly stimulate tectomanganate formation through enzymatic Mn(II) oxidation. However, some studies have reported direct tectomanganate formation from biogenic Mn oxide processes. Webb et al. [48], for example, reported pseudo-tunnel structures (todorokite-like), with $U^{VI}O_2^{2+}$ serving as a template ion, during Mn oxide biogenesis by *Bacillus* sp. SG-1. In addition, Saratovsky et al. [49] reported todorokite-like biogenic Mn oxides from *Acremonium* KR21-2 in solid agar media.

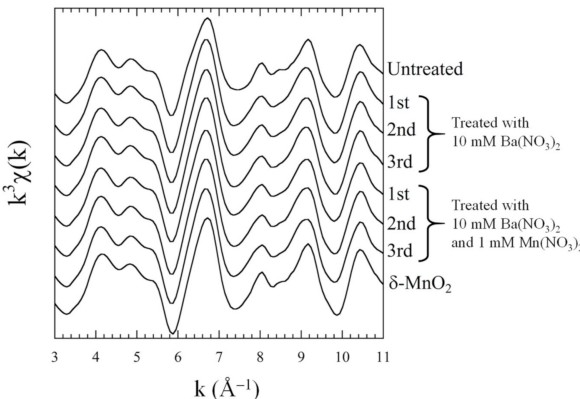

**Figure 3.** Mn K-edge extended X-ray absorption fine structure spectra for the newly formed biogenic Mn oxides (BMOs) treated with and without 10 mM $Ba(NO_3)_2$ and 1 mM $Mn(NO_3)_2$. δ-$MnO_2$ was plotted as a reference phyllomanganate for comparison.

The XRD patterns of the newly formed BMOs repeatedly treated in $Ba^{2+}$ and exogenous $Mn^{2+}$ are typical of birnessite (Figure 4). As the initial $Ba^{2+}$ concentration increased, the peak intensities for the (001) and (002) basal reflections also significantly increased, especially the (002) reflection peak. The full width at half maximum (FWHM) of basal reflections for the resulting BMOs narrowed in comparison to those of the newly formed BMOs treated with exogenous $Mn^{2+}$ without $Ba^{2+}$, indicating tighter layer stacking as the irreversible $Ba^{2+}$ content increased. In addition to Mn K-edge EXAFS results (see above), the XRD results confirmed minor alteration from turbostratic to tightly stacked (well-ordered along the *c*-axis) birnessite, with subsequent irreversible $Ba^{2+}$ accommodation, possibly into the interlayer space. This observation is consistent with the absence of alteration in the diffractogram for exchangeable $Ba^{2+}$ removal using the Cu(II) procedure (Figure S4).

Without exogenous $Mn^{2+}$, the repeated treatment in $Ba^{2+}$ solutions at 1 and 10 mM significantly weakened the (001) and (002) basal reflection peaks (Figure S5), suggesting that $Ba^{2+}$ sequestration on the "preformed" BMOs increased the disorder in its turbostratic structure along the *c*-axis hosting most reversible $Ba^{2+}$. Xhaxhiu [50] demonstrated that a chemically synthesized turbostratic $Na^+$–birnessite readily changes to phyllomanganate with disorder along the *c*-axis after treatment in a $Ba^{2+}$ solution. In addition to the loss of the (001) and (002) peaks by the heated (enzymatically inactivated) BMOs upon treatment in $Ba^{2+}$, even with exogenous $Mn^{2+}$ (Figure 4), we conclude that active Mn oxide formation and coexistence with $Ba^{2+}$ are prerequisites for producing tightly stacked birnessite sheets, with irreversible $Ba^{2+}$ incorporation in the interlayer.

Analysis of the Ba K-edge EXAFS data strongly supports the irreversible $Ba^{2+}$ incorporation into the interlayer of the tight birnessite structure. The newly formed BMOs treated thrice in 10 mM $Ba^{2+}$ with and without exogenous $Mn^{2+}$ (1 mM) displayed similar EXAFS

oscillations (Figure 5), with their radial structural functions (RSFs) indicating Ba–O shells at R + ΔR = 2.1 Å. The newly formed BMOs treated with $Ba^{2+}$ and exogenous $Mn^{2+}$ also exhibited small peaks attributed to the Ba–Mn scattering path at R + ΔR = 3.6 Å. The second shell of Ba–Mn scattering was clearer after the extraction using a 10 mM $Cu(NO_3)_2$ solution for removing exchangeable (reversible) $Ba^{2+}$. This indicates that the irreversible $Ba^{2+}$ on the BMOs created an inner-sphere complex in association with dehydration. The complex suggests covalent bonding of $Ba^{2+}$ to the oxygen atoms of the $MnO_6$ octahedra at interlayer sites. This strong $Ba^{2+}$ bonding is probably irreversible, and it stimulated the structure development of tightly stacked birnessite. Further studies are needed to clarify the atomic-level $Ba^{2+}$ incorporation mechanism during enzymatic Mn(II) oxide formation.

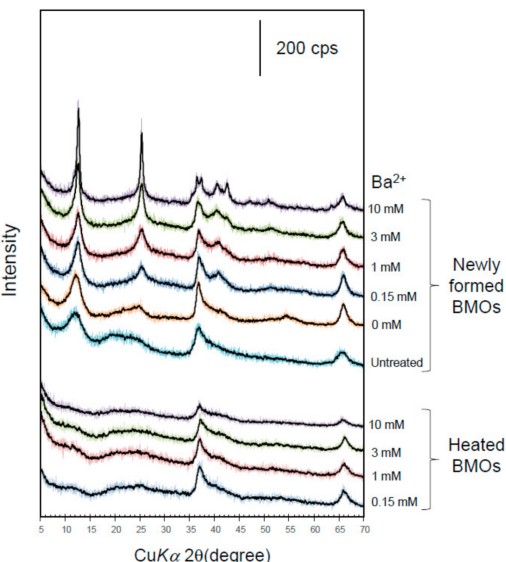

**Figure 4.** X-ray diffractograms from analysis of the newly formed and heated biogenic manganese oxides (1 mM as Mn) treated thrice with mixtures of $Ba(NO_3)_2$ (0–10 mM) and 1 mM $Mn(NO_3)_2$ in 20 mM HEPES (pH 7.0). The bathing solutions were renewed every 24 h.

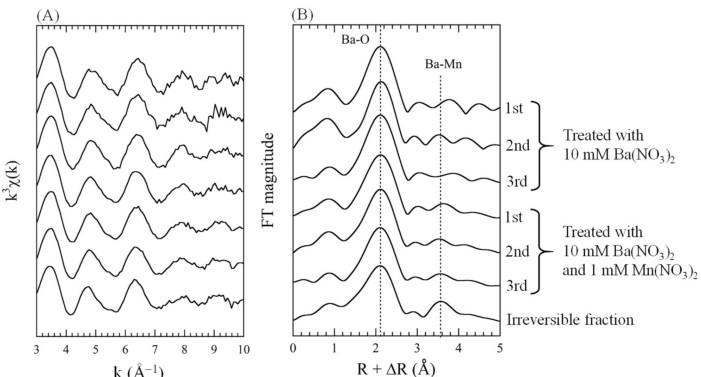

**Figure 5.** Ba K-edge EXAFS spectra of the newly formed BMOs treated in 10 mM $Ba(NO_3)_2$ with and without 1 mM $Mn(NO_3)_2$, highlighting the (**A**) EXAFS oscillations and (**B**) corresponding radial structural functions (RSFs). The irreversible fraction indicates the $Ba^{2+}$ left on the BMO after extracting the reversible $Ba^{2+}$ fraction using 10 mM $Cu(NO_3)_2$.

### 3.4. $Sr^{2+}$, $Ca^{2+}$, and $Mg^{2+}$ Sequestration by Newly Formed or Heated BMOs Involving Exogenous $Mn^{2+}$

To determine if irreversible sequestration during active Mn oxide formation is specific for $Ba^{2+}$ or if it is possible for other alkaline earth metal ions, we treated newly formed BMOs (1 mM Mn) thrice in 10 mM $Sr^{2+}$, $Ca^{2+}$, or $Mg^{2+}$, including exogenous 1 mM $Mn^{2+}$ (20 mM HEPES at pH of 7.0). For all alkaline earth metal ions, the exogenous $Mn^{2+}$

was converted to solid Mn with an efficiency > 97% after every 24 h, with reducible Mn exceeding 89%, confirming retention of the Mn(II) oxidation efficiency (Figure 6). The $Sr^{2+}$ sequestered was $0.31 \pm 0.01$, $0.40 \pm 0.05$, and $0.48 \pm 0.05$ mM after the first, second, and third treatments, respectively, with exchangeable (reversible) $Sr^{2+}$ exceeding 98% (Figure 6 and Table S2). Exchangeable $Ca^{2+}$ also dominated the sequestered $Ca^{2+}$ (>94.5%) with the totals ($0.28 \pm 0.05$, $0.37 \pm 0.01$, and $0.51 \pm 0.03$ mM) close to those for $Sr^{2+}$ (Figure 6 and Table S2). However, the sequestered quantities of $Mg^{2+}$ ($0.18 \pm 0.03$, $0.37 \pm 0.02$, and $0.41 \pm 0.01$ mM, respectively) were lower, with a higher exchangeable fraction of >82% (Figure 6 and Table S2). These results indicate that the sequestration of these ions on the newly formed BMOs is controlled primarily by reversible sorption, even with simultaneous exogenous $Mn^{2+}$ oxidation. In addition, all BMOs produce XRD patterns typical of turbostratic birnessite, with the (001) and (002) basal peaks broader than those of the tightly stacked birnessite-type BMOs involving irreversible $Ba^{2+}$ (Figure 7). Therefore, the high irreversible sequestration appears to be limited to $Ba^{2+}$.

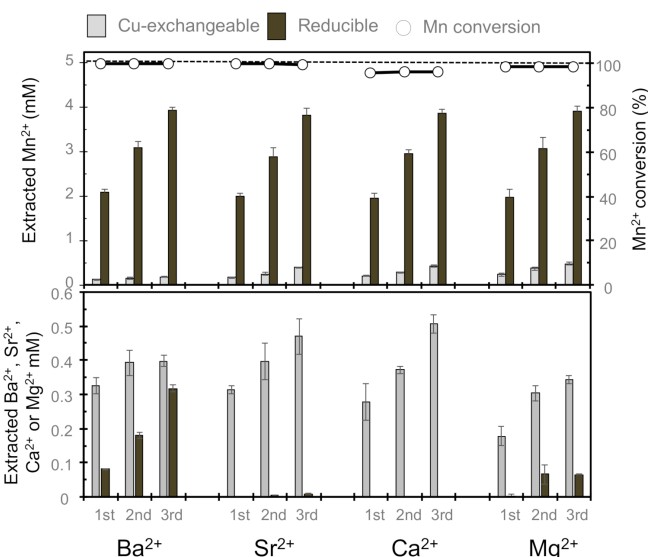

**Figure 6.** Illustration of the two-step extraction of $Ba^{2+}$ and Mn from the newly formed biogenic manganese oxides during repeated treatment with mixtures of 10 mM $Ba(NO_3)_2$, $Sr(NO_3)_2$, $Ca(NO_3)_2$, or $Mg(NO_3)_2$ with 1 mM $Mn(NO_3)_2$ in 20 mM HEPES (pH of 7.0). The bathing solutions were renewed thrice every 24 h. The conversion (%) from $Mn^{2+}$ to solid Mn is displayed in the top panel.

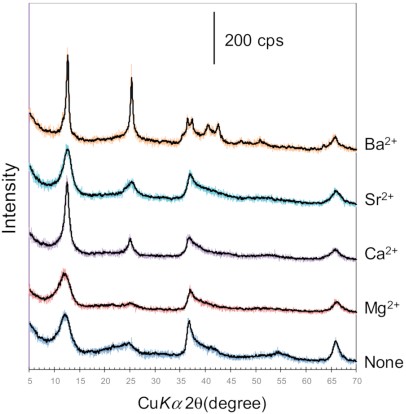

**Figure 7.** X-ray diffraction analysis of newly formed biogenic manganese oxides (1 mM as Mn) with mixed solutions of 10 mM $Ba(NO_3)_2$, $Sr(NO_3)_2$, $C(NO_3)_2$, or $Mg(NO_3)_2$ with 1 mM $Mn(NO_3)_2$ in 20 mM HEPES (pH 7.0). Bathing solutions were renewed thrice every 24 h.

### 3.5. Active $Mn^{2+}$ Oxidation Sequestration Selectivity Enhancement for $Ba^{2+}$

To assess the sequestration selectivity among the alkaline earth metal ions, we conducted competitive sequestration experiments in a solution containing 1 mM $Ba^{2+}$, 1 mM $Sr^{2+}$, and 1 mM exogenous $Mn^{2+}$ (20 mM HEPES at pH 7.0) using the newly formed BMOs (1 mM Mn). After three treatments, with renewal of the bathing solution every 24 h, the exogenous $Mn^{2+}$ was converted to the solid phase (>99% efficiency), with reducible Mn dominating (>92.0 ± 0.5%; Figure 8), indicating active Mn oxide formation. The sequestration efficiency for $Ba^{2+}$ was 24.6 ± 1.2%, 12.5 ± 0.3%, and 10.4 ± 0.9% for the first, second, and third treatment (cumulative efficiency 15.8 ± 0.8%), respectively (Table S3). The sequestration efficiencies of 5.5 ± 0.4%, <1%, and <1% (cumulative ~0.7%), respectively, for $Sr^{2+}$ were significantly lower. The two-step extraction produced $Ba^{2+}_{seq}/Sr^{2+}_{seq}$ molar ratios for the resultant BMO phase ranging from 4.9 to 9.1 (Figure 8D), highlighting a clear increase with renewal of the bathing solution. Similar trends were observed for $Ba^{2+}/Ca^{2+}$ and $Ba^{2+}/Mg^{2+}$ with exogenous $Mn^{2+}$, with $Ba^{2+}_{seq}/Ca^{2+}_{seq}$ and $Ba^{2+}_{seq}/Mg^{2+}_{seq}$ molar ratios increasing from 7.1 to 10.6 and from 13.1 to 31.2, respectively (Figure 8D). Evidently, even in the competitive sequestration experiments, the proportion of irreversible (reducible) $Ba^{2+}$ significantly increased as the exogenous $Mn^{2+}$ progressed (Figure 8B), while the coexisting $Sr^{2+}$, $Ca^{2+}$, and $Mg^{2+}$ were sequestered mostly as exchangeable fractions (Figure 8C). For example, the reducible $Ba^{2+}$ increased from 26.6 ± 1.5% to 40.3 ± 0.3% and then to 46.8 ± 0.3%, while exchangeable $Sr^{2+}$ dominated the sequestered $Sr^{2+}$ (>99%) throughout the experiment (Table S3).

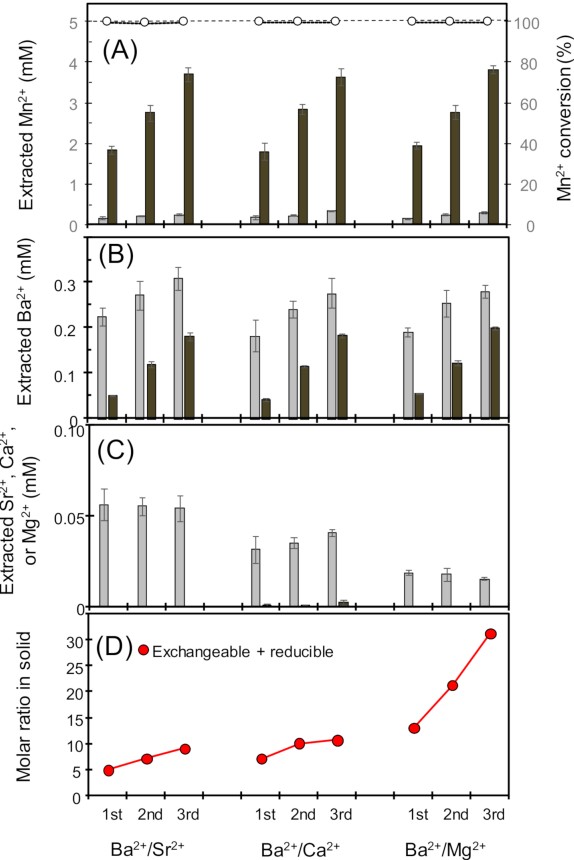

**Figure 8.** Diagram showing the two-step extraction of (**A**) Mn; (**B**) Ba; and (**C**) Sr, Ca, or Mg from the newly formed biogenic manganese oxides during repeated treatments with mixed solutions of 1 mM $Ba(NO_3)_2$ and 1 mM $Mn(NO_3)_2$ in 20 mM HEPES (pH of 7.0) with 1 mM $Sr(NO_3)_2$, $Ca(NO_3)_2$, or $Mg(NO_3)_2$. The bathing solutions were renewed thrice every 24 h. (**D**) Plot of the Ba/Sr, Ba/Ca, and Ba/Sr molar ratios in the solid phases demonstrating sequestration selectivity.

Competitive sequestration experiments without exogenous $Mn^{2+}$ also revealed that exchangeable $Ba^{2+}$ was dominant in the sequestered $Ba^{2+}$ (>94.4%) (Figure 9B and Table S3), consistent with the $Ba^{2+}$ restricted sequestration experiments. Most $Sr^{2+}$, $Ca^{2+}$, and $Mg^{2+}$ sequestered were also in the exchangeable fractions (Figure 9C). The $Ba^{2+}_{seq}/Sr^{2+}_{seq}$, $Ba^{2+}_{seq}/Ca^{2+}_{seq}$, and $Ba^{2+}_{seq}/Mg^{2+}_{seq}$ molar ratios were, however, lower than those with exogenous $Mn^{2+}$, ranging from 4.3 to 5.1, 4.1 to 5.1, and 5.7 to 6.5, respectively (Figure 9D). Consequently, the reversible $Ba^{2+}$ sequestration by the preformed BMOs caused lower $Ba^{2+}$ selectivity compared to the irreversible $Ba^{2+}$ sequestration into tightly stacked birnessite-type BMOs.

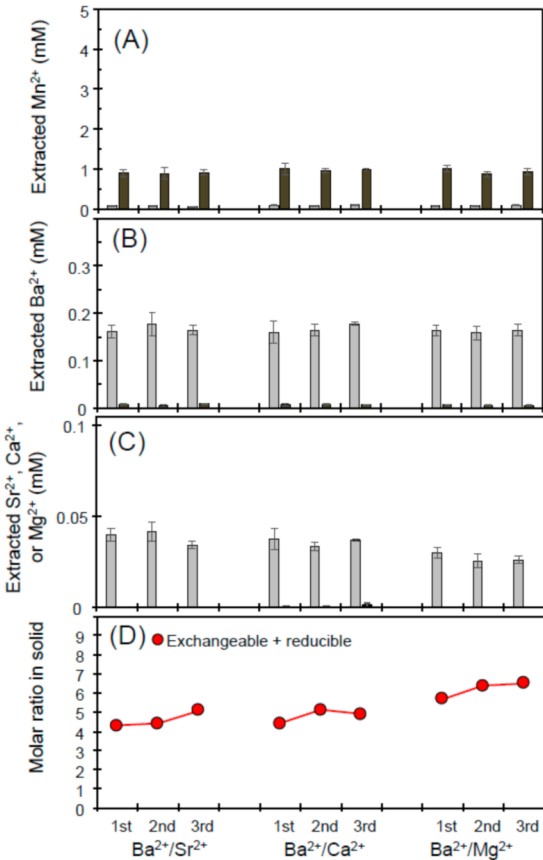

**Figure 9.** Diagram showing the two-step extraction of (**A**) Mn; (**B**) Ba; and (**C**) Sr, Ca, or Mg from the newly formed biogenic manganese oxides during repeated treatment with mixtures of 1 mM $Ba(NO_3)_2$ and 1 mM $Sr(NO_3)_2$, $Ca(NO_3)_2$, or $Mg(NO_3)_2$ in 20 mM HEPES (pH of 7.0) without exogenous $Mn^{2+}$. The bathing solutions were renewed thrice every 24 h. (**D**) Plot of the Ba/Sr, Ba/Ca, and Ba/Sr molar ratios in the exchangeable and solid (exchangeable + reducible) phases showing sequestration selectivity.

The XRD patterns of the newly formed BMOs treated thrice in a mixture of 1 mM $Ba^{2+}$ and 1 mM $Sr^{2+}$ resembled those for BMOs treated in the 1 mM $Ba^{2+}$ solution more than those for BMOs treated in the 1 mM $Sr^{2+}$ solution (Figure S6). This confirms that only $Ba^{2+}$ was irreversibly incorporated, even in the competitive sequestration experiments. The biological $Ba^{2+}$ sequestration process is likely in environments with simultaneous supply of $Mn^{2+}$ and $Ba^{2+}$, which subsequently contributes to $Ba^{2+}$ accumulation in birnessite-type Mn oxides.

## 4. Conclusions

In this study, we present results showing that irreversible $Ba^{2+}$ sequestration predominates during simultaneous enzymatic Mn oxidation. This process is a likely pathway for

$Ba^{2+}$ sequestration into naturally occurring Mn oxide phases, with microbial (enzymatic) activity occasionally catalyzing the process in the environment. Irreversible sequestration was limited to $Ba^{2+}$, with $Sr^{2+}$, $Ca^{2+}$, and $Mg^{2+}$ characterized by reversible sequestration, thereby explaining the preferential accumulation of $Ba^{2+}$ in Mn oxide phases in the environment. These findings improve understanding of the role of biogenic Mn oxidation in natural $Ba^{2+}$ cycling, particularly under conditions in which microbial Mn(II) oxidation dominates abiotic processes. The insights from this study also highlight the potential of enzymatically active BMOs for scavenging $Ba^{2+}$ from contaminated wastewaters.

**Supplementary Materials:** The following are available online at https://www.mdpi.com/2075-163X/11/1/53/s1: Figure S1: $Mn^{2+}$ oxidation by newly formed BMOs in $Mn(NO_3)_2$. Figure S2: XRD patterns of newly formed BMOs treated with $Mn(NO_3)_2$. Figure S3: Repeated treatment of newly formed BMOs $Ba(NO_3)_2$. Figure S4: Effect of $Cu^{2+}$-extraction on XRD patterns of newly formed and heated BMOs. Figure S5: XRD patterns of newly formed BMOs treated with $Ba(NO_3)_2$. Figure S6: XRD patterns of newly formed BMOs treated with mixed solutions of $Ba(NO_3)_2$ and $Sr(NO_3)_2$. Table S1: Data summary of sequestration experiments for $Ba^{2+}$. Table S2: Data summary of sequestration experiments for $Sr^{2+}$, $Ca^{2+}$, or $Mg^{2+}$. Table S3: Data summary of competitive sequestration experiments.

**Author Contributions:** Conceptualization, Y.T. and N.M.; methodology, Y.T., K.T. and N.M.; validation, Y.T., S.K., J.C., K.T. and N.M.; formal analysis, S.K. and J.C.; investigation, Y.T., K.T. and N.M.; data curation, Y.T., S.K. and J.C.; writing—original draft preparation, Y.T.; writing—review and editing, Y.T., K.T. and N.M.; visualization, Y.T.; supervision, Y.T.; project administration, Y.T.; funding acquisition, Y.T. All authors have read and agreed to the published version of the manuscript.

**Funding:** This research was funded by the Japan Society of the Promotion of Science, JSPS KAKENHI, no. 20K12222 (Y.T.).

**Institutional Review Board Statement:** Not applicable.

**Informed Consent Statement:** Not applicable.

**Data Availability Statement:** The data presented in this study are available in the Supplementary Materials (Tables S1–S3).

**Acknowledgments:** We would like to thank Editage for the editing assistance. The EXAFS measurements were performed with the approval of the Photon Factory, KEK (proposal no. 2018G111) and SPring-8 (proposal no. 2018B1012).

**Conflicts of Interest:** The authors declare no conflict of interest.

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
