# Peer review of "Preferential Elimination of Ba2+ through Irreversible Biogenic Manganese Oxide Sequestration"

_minerals, doi:10.3390/min11010053_

Round 1

Reviewer 1 Report

The manuscript investigates Ba2+ sequestration using fungal (biogenic) manganese oxides (BMOs). The authors show that exogenous Mn2+ is oxidized by the BMOs and form additional BMO. Selective irreversible sequestration of Ba2+ was demonstrated and confirmed by the formation of tightly stacked birnessite (with Ba2+ incorporated in to its interlayers). Other tested ions (Sr2+, Ca2+, and Mg2+) were sequestered reversibly and the original (turbostratic) structure of birnessite was preserved.

As the authors highlight, (i) the findings contribute to our understanding of the role of biogenic Mn oxidation in Ba-cycling, and (ii) enzymatically active BMOs could be utilized for eliminating Ba2+ from contaminated wastewaters.

The study presents interesting findings on biogenic BMO formation and Ba2+ sequestration. The experiments were well designed, and I find all parts of the manuscript well presented.

Detailed Comments:

Ln. 27: There is a surplus full stop in “.. Ba2+ from contaminated wastewaters..”. Please delete one.

Ln. 32: It might be better to use “elevated” or “increased” rather than “aggravated”.

Ln. 47 – 48: There is a redundant comma (the last one) in “..Zn2+, Ni2+, Co2+, and Pb2+, from aquatic environments because of its high sequestration affinity and capacity..”. Also, “its” in the sentence refers to BMOs, and I would recommend that the part of the sentence is changed to e.g.: “..Zn2+, Ni2+, Co2+, and Pb2+ from aquatic environments because of the high sequestration affinity and capacity of BMOs.”.

Ln. 58 – 59: I believe that the following: “..BMOs exhibit a potential for continuously remediating contaminated wastewaters and recovering metal ions.” might be better if rephrased to e.g. “..BMOs exhibit a potential for a continuous remediation of contaminated wastewaters as well as metal recovery.”

Ln. 64 – 66: Would a rephrasing of the following: “The results of this study demonstrate the application potential of enzymatically active BMOs for eliminating Ba2+ from contaminated wastewaters.” to: “The results of this study demonstrate the potential application of enzymatically active BMOs for Ba2+ removal from contaminated wastewaters.” sound better?

Ln. 87: What do the authors mean by “samples were corrected”? Could you please elaborate?

Ln. 114: Could the authors explain what “BN” means? Thank you.

Ln. 126: The only mention of to what statistical measures the authors refer is on ln. 165 (“The data shown represent the mean ± standard deviation (n = 3).”) and it might be worth defining that also in the main text, when such format occurs for the first time (ln. 126: “98.7 ± 0.1%”). Note: I presume the data are presented in a consistent way and refer to similar measures throughout the manuscript.

Ln. 206: There is a superscript missing in “Ba2+”.

Author Response

Thank you for your kind review on our manuscript. We completely revised the original manuscript based on comments and suggestions of Reviewer 1 as follows:

Comments/Suggestions: Ln. 27: There is a surplus full stop in “.. Ba2+ from contaminated wastewaters..”. Please delete one.

Response: We revised as suggested.

Comments/Suggestions: Ln. 32: It might be better to use “elevated” or “increased” rather than “aggravated”.

Response: We revised this word to “elevated”.

Comments/Suggestions: Ln. 47 – 48: There is a redundant comma (the last one) in “..Zn2+, Ni2+, Co2+, and Pb2+, from aquatic environments because of its high sequestration affinity and capacity..”. Also, “its” in the sentence refers to BMOs, and I would recommend that the part of the sentence is changed to e.g.: “..Zn2+, Ni2+, Co2+, and Pb2+ from aquatic environments because of the high sequestration affinity and capacity of BMOs.”.

Response: We revised as suggested.

Comments/Suggestions: Ln. 58 – 59: I believe that the following: “..BMOs exhibit a potential for continuously remediating contaminated wastewaters and recovering metal ions.” might be better if rephrased to e.g. “..BMOs exhibit a potential for a continuous remediation of contaminated wastewaters as well as metal recovery.”

Response: We revised as suggested.

Comments/Suggestions: Ln. 64 – 66: Would a rephrasing of the following: “The results of this study demonstrate the application potential of enzymatically active BMOs for eliminating Ba2+ from contaminated wastewaters.” to: “The results of this study demonstrate the potential application of enzymatically active BMOs for Ba2+ removal from contaminated wastewaters.” sound better?

Response: We revised as suggested.

Comments/Suggestions: Ln. 87: What do the authors mean by “samples were corrected”? Could you please elaborate?

Response: We apologize a spelling error. We revised to “samples were collected”

Comments/Suggestions: Ln. 114: Could the authors explain what “BN” means? Thank you.

Response: We revised “BN” to “boron nitride (BN)”.

Comments/Suggestions: Ln. 126: The only mention of to what statistical measures the authors refer is on ln. 165 (“The data shown represent the mean ± standard deviation (n = 3).”) and it might be worth defining that also in the main text, when such format occurs for the first time (ln. 126: “98.7 ± 0.1%”). Note: I presume the data are presented in a consistent way and refer to similar measures throughout the manuscript.

Response: We added the sentence “All the sequestration and extraction experiments were conducted in triplicated (n = 3), and data in the figures and tables are shown as mean ± standard deviation.” In Experimental Section. We deleted the sentence “The data shown represent the mean ± standard deviation (n = 3).” from Figure Caption in Figs 1 and 2.

Comments/Suggestions: Ln. 206: There is a superscript missing in “Ba2+”

Response: We revised as suggested.

Reviewer 2 Report

This paper considers irreversible preferential Ba2+ sequestration by biogenic manganese oxides from aquatic environments. This is a high quality experimental paper with a number of methods applied for the mineral structure characterization in details. Electron microscopic images would be also rather useful, but not necessary in this study. It is also suggested to  discuss the possible synthesis pathways of hollandite, romanechite and birnessite from the crystallographic and geochemical positions with the corresponding formulas and reactions given in the supplementary information. The results obtained in this work are of significant interest both for fundamental biogeochemistry, as well as for applied ecological studies and wastewater remediation. Thus, this paper is worth publishing in the journal Minerals in the present form.

Author Response

Thank you for your kind review on our manuscript. We revised the original manuscript based on the comments and suggestions by Reviewer 2.